

# Should the frequency, severity, or both response scales be used for multi-item dental patient-reported outcome measures?

Swaha Pattanaik[1], Mike T. John[1], Seungwon Chung[2] and San Keller[3]

[1] Department of Diagnostic and Biological Sciences, School of Dentistry, University of Minnesota, Minneapolis, MN, United States of America

[2] Department of Educational Psychology, College of Human Development, University of Minnesota, Minneapolis, MN, United States of America

[3] American Institutes of Research, Chapel Hill, NC, United States of America

Corresponding author
Swaha Pattanaik,
swahapattanaik@gmail.com

## ABSTRACT

**Background**. The Oral Impacts on Daily Performances (OIDP) index asks the respondents to indicate both, the frequency and severity of the impact. However, it is not clear if the two scaling methods are correlated, and if using one scale is sufficient. The purpose of the study was to investigate the correlation between frequency and severity rating scales of the OIDP instrument, and whether only one of the rating scales can be used instead of both.

**Methods**. A battery of patient-reported outcome questionnaires were administered to a consecutive sample of adult dental patients from HealthPartners dental clinics in Minnesota ($N = 2,115$). Only those who responded to any of the OIDP items were included in the analysis for this study ($N = 873$). We assessed correlations between the frequency and severity scales for all OIDP items, and for the summary scores of the two OIDP response scales. We additionally fit a categorical structural equation model (SEM) (or an item factor analysis model) and examined the correlation between two latent variables (Frequency and Severity).

**Results**. The correlation estimates for all OIDP items were greater than 0.50, indicating large correlations between the frequency and severity scores for each OIDP item. The correlation estimate between the two summary scores was 0.85 (95% CI [0.82–0.86]). When we calculated the correlation coefficient using a latent variable model, the value increased to 0.96 (95% CI [0.93–1.00]).

**Conclusion**. Our study findings show that OIDP frequency and severity scores are highly correlated, which indicates the use of one scale only. Based on previous evidence, we recommend applying the frequency rating scale only in research and clinical settings.

## BACKGROUND

A dental patient-reported outcome measure (dPROM) is an instrument, questionnaire, scale, or survey that captures patient-reported outcomes (dPROs) for adult dental patients and expresses it numerically in a score (*John, 2018*; *Paulson et al., 2021*). The numerical score captures the oral diseases' impact (*John, 2018*; *Paulson et al., 2021*). Oral health-related quality of life (OHRQoL) is arguably the most important dPRO as it measures the impact of oral diseases on people and their well-being and life quality (*Paulson et al., 2021*; *Mittal et al., 2019*). Consequently, several dPROMs are available to assess OHRQoL and they use various response scales to characterize the impact (*Mittal et al., 2019*). Frequency and severity are the two most common ways in which dPROMs assess the quality of the impact (*Paulson et al., 2021*; *Reissmann, 2021*). While dPROMs such as the Oral Health Impact Profile (OHIP) assess the temporal quality of an impact such as frequency, that is, how often the impact occurred (*Slade & Spencer, 1994*), other dPROMs such as the Orofacial Esthetic Scale (OES) assess intensity or severity of the impact (*Larsson et al., 2010*). Overall, researchers seem to agree that an ordinal response scale, *i.e.,* several steps on a frequency or severity spectrum, is preferred to a dichotomous response scale, *e.g.*, yes-no: an ordinal response provides more variability in the score distribution and hence leads to greater score precision. However, which feature of the impact, whether frequency or severity, is better, is not clear.

Theoretically, assessing both frequency and severity would be more comprehensive than assessing either alone; yet, assessing both increases administrative and patient burden. The patients have to recall not only the frequency of the impact but also its severity. This requires more cognitive processing. For instance, the patients need to decide what severity to report, the severity they are experiencing in the moment, average intensity over a certain period of time, or the highest intensity in this period (*Reissmann, 2021*). Research shows that respondents find remembering details of events from the previous day difficult, and thus averaging intensity over even a day may be quite cognitively demanding (*Organisation for Economic Co-operation and Development, 2013*). Reissmann recommends using one response format for each item and not changing response formats between items (*Reissmann, 2021*).

Given the burden and complications involved in including two response scales for each impact, we sought to determine the degree to which the two response scales provided redundant information. If the information they provide is redundant, it would be possible to use just one response scale and so to ameliorate the problems associated with dual response scales. The Oral Impacts on Daily Performances (OIDP) index, a widely used socio-dental indicator, is unique in that it quantifies oral impacts by asking respondents to indicate both the frequency and severity of the impacts (*Adulyanon & Sheiham, 1997*). To answer the question of whether impact frequency or impact severity are redundant, existing OIDP data offer a natural, cost-effective opportunity. If scores based on frequency responses would not be highly correlated with those based on severity responses, we would assume that both response scales are required because they measure different things. If
frequency and severity were highly correlated, they measure the same thing, and only one quality is necessary in future studies.

Results of such a study comparing frequency and severity of oral health impacts would have both narrow and broad implications. From a narrow point of view, using a single response scale (that is, either frequency or severity but not both) for the OIDP would help reduce costs and time to administer the instrument. From a broader point of view, these findings would advance our understanding of how to design the response format for dPROMs in general. For example, findings for the OIDP can likely be generalized to other OHRQoL instruments, as OIDP is highly correlated with other OHRQoL instruments (*Robinson et al., 2003*; *John et al., 2016*). Despite these remarkable practical implications, we are aware of no other studies that have investigated if the OIDP, or any other dPROMs, require both frequency and severity response scales. The purpose of the study was to investigate the correlation between frequency and severity rating scales of the OIDP instrument, and whether only one of the rating scales can be used instead of both.

## METHODS

### Participants, study design and setting

This is a secondary research study using data from another cross-sectional study (*Simancas-Pallares et al., 2018*; *Simancas-Pallares et al., 2020*). The researchers in the original study used a consecutive sampling strategy to recruit a total of 2,115 English—and Spanish— speaking adult dental patients from HealthPartners dental clinics in Minnesota, USA, between July 2014, and April 2016. A battery of dPROMs consisting of items from the OIDP, Oral Health Impact Profile (OHIP), and General Oral Health Assessment Index (GOHAI), were administered to the participants. They filled out the self-administered questionnaires at home and sent them back to the researchers. All participants received $50 incentives for their participation. Please refer to the previous research studies (*Simancas-Pallares et al., 2018*; *Simancas-Pallares et al., 2020*) for further details about data collection and recruitment. This research was conducted in accordance with accepted ethical standards for human-patient research practice, undergoing review and approval by the Institutional Review Board of the HealthPartners Institute in Minneapolis, MN (registration A11-136). All the participants completed an informed consent form before their enrollment.

### Measure: OIDP

The OIDP is a widely used socio-dental indicator, that measures the impact of poor oral health on individuals' performance of physical, psychological, and social performances. *Adulyanon & Sheiham (1997)* originally developed the OIDP in the adult general population in Thailand. Since then, psychometric soundness of the OIDP has been supported in different countries (*Åstrøm & Okullo, 2003*; *Astrøm et al., 2005*; *Dorri, Sheiham & Tsakos, 2007*; *Erić et al., 2012*; *Jung et al., 2008*; *Kida et al., 2006*; *Lajnert et al., 2016*; *Lawal, Taiwo & Arowojolu, 2013*; *Montero, Bravo & Albaladejo, 2008*; *Naito et al., 2007*; *Ostberg, Andersson & Hakeberg, 2008*; *Purohit et al., 2012*). There are eight items or daily performances on the OIDP. See Table 1 for the list of daily performances assessed in the OIDP. There are eight items or questions asked of respondents about whether their

**Table 1   Performances assessed in the Oral Impacts on Daily Performances.**

| List of daily performances |
| --- |
| **Physical Performances** |
|     a. Eating and enjoying food |
|     b. Speaking and pronouncing clearly |
|     c. Cleaning teeth |
| **Psychological Performances** |
|     d. Sleeping and relaxing |
|     e. Smiling, laughing, and showing teeth without embarrassment |
|     f. Maintain usual emotional state without being irritable |
| **Social Performances** |
|     g. Carrying out major work or social role |
|     h. Enjoying contact with people |

teeth, mouth or denture caused them difficulty with any of the eight daily performances within the last 6 months. Therefore, OHRQoL is a negative concept, and the sum scores will be problem indices. If respondents indicate ''yes'' to problems with teeth, mouth or denture having an impact on any of the daily performances, then they answer questions about the frequency and severity of the impact of dental problems on that performance. Table 2 shows specific frequency and severity questions corresponding to each OIDP performance. Both frequency and severity scores are derived separately. The last row in Table 2 shows overall scoring method for the OIDP and Table 3 shows criteria of frequency score.

As shown in Table 2 under #2 with specific criteria seen in Table 3, the OIDP provides respondents with two ways to report frequency with which problems associated with mouth, teeth, or denture impact on a particular performance. If the impact occurs regularly, respondents report on the frequency over the past six months ranging from ''Never'' to ''Every or nearly every day''. If the impact was restricted to a period of time (*i.e.,* a spell), respondents report on how many days they suffered the impact in the past 6 months ranging from ''0'' to more than 3 months. As shown in Table 2 under #3 the perceived severity of the impact of problems with mouth, teeth, or dentures on any of the eight performances was assessed by asking the respondents to rate the severity of the impact on a scale from 0 to 5, where 0 = ''no effect'' and 5 = ''very severe effect''. Then, an individual score for the impact of problems with teeth, mouth or dentures on each performance is calculated by multiplying the value for frequency with the value for severity. The total score is the sum of all individual scores divided by the maximum possible score (*e.g.,* 8 performances*5 frequency scores*5 severity score = 200) (*Slade & Spencer, 1994*; *Adulyanon & Sheiham, 1997*). See the last row of Table 2 for the OIDP scoring method.

## Correlation analysis

We assessed correlations between the frequency and severity rating scales for: (a) all OIDP items, (b) OIDP sum scores, and (c) severity and frequency scales as latent variables.

**Table 2 Severity- and frequency-related questions and the scoring method in the Oral Impacts on Daily Performances.**

**OIDP questions asked for each daily performance or item**

(1) **In the past 6 months, have problems with mouth, teeth, or dentures caused you any difficulty in performance (see Table 1 for the list of performances)?**

*If yes,*

    (2a) Have you had this difficulty in (performance) on a regular/periodic basis or for a period/spell?

    *-If ability restricted on a "regular/periodic basis",*

        (2b). During the past six months how often have you had this difficulty?[b]

    *-or, if ability restricted "on a period/spell"*

        (2c). For how much of the past 6 months have you had this difficulty?[b]

(3) **And using a scale from 0 to 5, where 0 is no effect and 5 is a very severe effect, which number would you say reflects what the difficulty in (performance) had on your daily life?**

**Scoring method for Oral Impact on Daily Performances**

OIDP score =

[(frequency score[b] of "Eating" × severity score[b] of "Eating") +

(frequency of "Speaking" × severity of "Speaking") +

(frequency of "Cleaning teeth" × severity of "Cleaning teeth") +

(frequency of "Sleeping" × severity of "Sleeping") +

(frequency of "Smiling" × severity of "Smiling") +

(frequency of "Emotional stability" × severity of "Emotional stability") +

(frequency of "Major role" × severity of "Major role") +

(frequency of "Contact with people" × severity of "Contact with people")]

× 100/200[c]

**Notes.**

[a]Scores range from 0 to 5.

[b]Please see Table 3 for responses and scores for answers to 2b and 2c.

[c]Maximum possible score [Sum of 8 performances score × 5 frequency score × 5 severity score] = 200.

**Table 3 Criteria of frequency score of affected performances over past six months.**

| Frequency (for people affected on a regular or periodic basis) | Duration (for people affected for a period/spell) | Score |
| --- | --- | --- |
| Never affected in past 6 months | 0 days | 0 |
| Less than once a month | Up to 5 days in total | 1 |
| Once or twice a month | Up to 15 days in total | 2 |
| Once or twice a week | Up to 30 days in total | 3 |
| 3–4 times a week | Up to 3 months in total | 4 |
| Every or nearly every day | Over 3 months in total | 5 |

## (1) Correlation analysis for OIDP items

We obtained polychoric correlation coefficients between each OIDP item's frequency and severity scales and their 95% confidence intervals. Polychoric correlation analysis is an appropriate correlation analysis method for ordinal data. We followed *Cohen*'s (*1988*) criteria (*Cohen, 1988*) to interpret the magnitude of correlation coefficients ($r = 0.10$, $r = 0.30$, and $r = 0.50$ represent small, moderate, and large magnitudes).

## (2) Correlation analysis for the sum scores

We also summed up the individual item severity and frequency scores to derive two separate total scores for the OIDP frequency and severity scales. We then computed the correlation

coefficient and its 95% confidence interval for these scores. In addition, we plotted the two total scores against each other to provide a visual representation of the strength of their relationship.

### (3) Correlation analysis using latent variable modeling

We fit a structural equation model (SEM) to the OIDP data. In SEM, we relate frequency and severity items to latent variables, which corrects for attenuation due to measurement error (*Bedeian, Day & Kelloway, 1997*). We estimated the correlation between the two latent variables and determined if the correlation is close to one using a threshold of 0.85 (*Clark & Watson, 1995*; *Kline, 2011*).

Specifically, we fit a categorical SEM or an item factor analysis model wherein diagonally weighted least squares (DWLS) estimation was applied when fitting the model to the polychoric correlation matrix. Since SEM is a model-based approach, we evaluated model fit using the chi-square test, root mean square error of approximation (RMSEA) and incremental fit indices including comparative fit index (CFI) and Tucker–Lewis index (TLI). We assessed model fit based on the commonly applied guidelines: RMSEA: $\leq$0.06; and CFI, TLI: $\geq$0.95 (*John et al., 2014*; *Hu & Bentler, 1999*).

## RESULTS

### Participant characteristics

Our sample consisted a total of 873 study participants. There were more females ($n = 545$) than males ($n = 328$) participating in the studies. The mean age of the participants was $53.03 \pm 15.71$ (range 22–97) years.

### Descriptive data for severity and frequency scores

A total of 835 study participants responded to all items in the severity scale (*i.e.,* observations without missing values in the severity scale), with a mean score of 8.04 and standard deviation of 8.12, while 773 participants responded to all items in the frequency scale (*i.e.,* observations without missing values in the frequency scale), with a mean score of 10.27 and standard deviation of 9.11. A total of 762 participants responded to all items in both severity and frequency scales and were thus considered in our analysis. We have summarized the prevalence of the response categories in Table 4.

### Correlation analysis

#### (1) Correlations analysis for OIDP items

In Table 5 we present correlation coefficients and confidence intervals for the OIDP items. Correlation coefficients for all items were above 0.50, indicating a strong or large correlation between the frequency and severity scores. The coefficients were associated with relatively narrow confidence intervals indicating precise estimates.

#### (2) Correlation analysis for the sum scores

The correlation estimate between the two summary scores was 0.85 (95% CI [0.82–0.86]), indicating a strong correlation between the two rating scales. The estimates were supplemented by a visual representation of the strong correlation between the sum scores (as shown in Fig. 1).

**Table 4  Prevalence of response category [%].**

| Response options | Severity item | | | | | | | | | Frequency item | | | | | | | |
|---|---|---|---|---|---|---|---|---|---|---|---|---|---|---|---|---|---|
| | 1 | 2 | 3 | 4 | 5 | 6 | 7 | 8 | | 1 | 2 | 3 | 4 | 5 | 6 | 7 | 8 |
| **0** | 31 | 64 | 40 | 56 | 39 | 63 | 72 | 65 | **0** | 22 | 58 | 34 | 49 | 33 | 56 | 65 | 58 |
| **1** | 25 | 15 | 24 | 18 | 19 | 15 | 11 | 12 | **1** | 19 | 10 | 19 | 15 | 14 | 13 | 11 | 11 |
| **2** | 13 | 6 | 12 | 9 | 11 | 7 | 5 | 6 | **2** | 15 | 7 | 13 | 11 | 11 | 7 | 5 | 7 |
| **3** | 15 | 6 | 12 | 7 | 11 | 6 | 4 | 6 | **3** | 12 | 5 | 9 | 6 | 9 | 6 | 3 | 5 |
| **4** | 7 | 4 | 5 | 4 | 7 | 3 | 3 | 4 | **4** | 10 | 5 | 7 | 6 | 6 | 4 | 3 | 4 |
| **5** | 6 | 2 | 5 | 2 | 11 | 2 | 2 | 4 | **5** | 14 | 7 | 13 | 5 | 19 | 4 | 3 | 6 |
| **Miss** | 2 | 3 | 2 | 3 | 2 | 4 | 4 | 4 | **Miss** | 6 | 9 | 6 | 8 | 7 | 10 | 10 | 9 |

**Table 5  Correlation coefficients and confidence intervals for OIDP items.**

| OIDP Items | Correlation (95% confidence interval) |
|---|---|
| *Item 1:* Eating and enjoying food | 0.75 (0.72–0.78) |
| *Item 2:* Speaking and pronouncing clearly | 0.82 (0.80–0.84) |
| *Item 3:* Cleaning teeth | 0.74 (0.71–0.78) |
| *Item 4:* Sleeping and relaxing | 0.82 (0.79–0.84) |
| *Item 5:* Smiling, laughing, and showing teeth without embarrassment | 0.83 (0.81–0.85) |
| *Item 6:* Maintain usual emotional state without being irritable | 0.74 (0.70–0.77) |
| *Item 7:* Carrying out major work or social role | 0.78 (0.75–0.81) |
| *Item 8:* Enjoying contact with people | 0.80 (0.77–0.82) |
| Summary score | 0.85 (0.82–0.86) |

***(3) Correlation analysis using latent variable modeling***

When we calculated the correlation coefficient in a latent variable model wherein the measurement error was considered, the value increased to 0.96 (95% CI [0.93–1.00]) (as shown in Fig. 2). Measurement error accounts for the difference between the correlation of 0.85 and 0.96. However, we note that the model fit to the data was marginal ($\chi^2(103) = 739.95$, $p < 0.001$, RMSEA = 0.090, CFI = 0.944, TLI = 0.935). This is not surprising given that the same item stem is used in severity and frequency scales. This was also evidenced in the modification indices. Six out of 8 items (item 1, 2, 3, 4, 5, and 8) showed MI values greater than 10.

## DISCUSSION

We found large correlations between severity and frequency scale scores for individual item scores and summary scores, presenting evidence of redundancy in the information provided by the two types of response formats. Dental researchers have previously used OIDP with only the frequency scale (*Åstrøm & Okullo, 2003*). *Adulyanon & Sheiham (1997)* investigated if multiplying the frequency score and severity score (that is, computing the traditional OIDP score) would add more information about impact than basing the scores on either frequency or severity responses alone. They applied three multiple regression

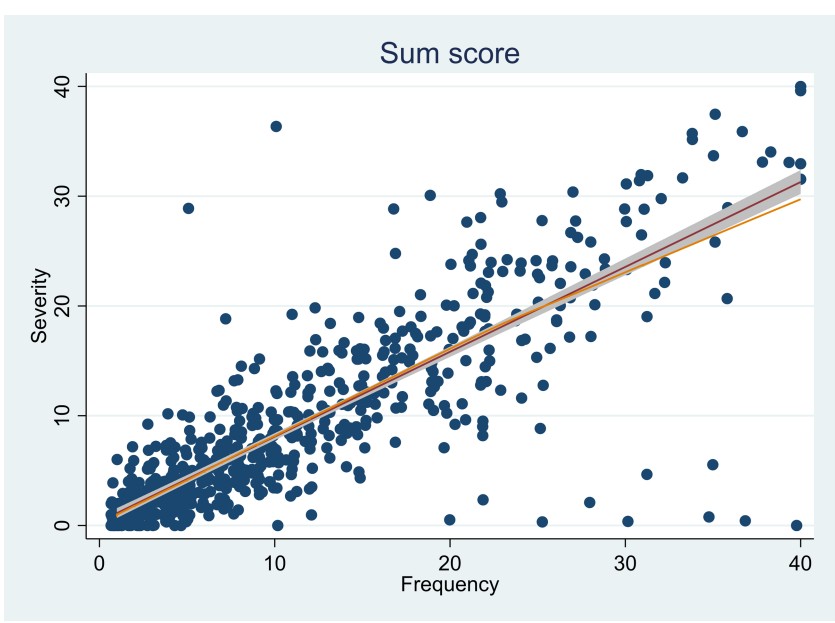

**Figure 1** Plot representing the correlation between the sum scores from the severity and frequency rating scales.

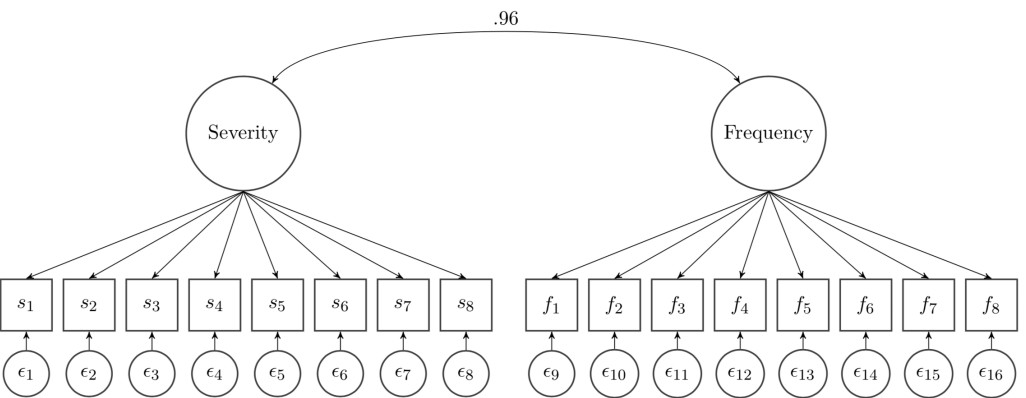

**Figure 2** Latent variable modeling between the two rating scales.

models to predict DMFT (*i.e.*, the sum of the number of permanent teeth that were decayed, missing due to caries, and filled) as well as the number of sextants with deep periodontal pockets. They observed that the improvement of OIDP compared to either frequency or severity score used alone was not statistically significant in the prediction test, hence they recommended using one of them for improving simplicity and efficiency. Frequency was suggested as a better representative single choice due to its better reproducibility compared with severity. Other researchers have also used only the OIDP frequency scores to assess OHRQoL in adolescent (*Åstrøm & Okullo, 2003*; *Shekhawat, Samuel & Chauhan, 2021*) and adult general (*Astrøm et al., 2005*) and dental patient populations (*Lawal, Taiwo &*

*Arowojolu, 2013*; *Lajnert et al., 2016*; *Ostberg, Andersson & Hakeberg, 2008*; *Gülcan et al., 2014*) across the world. Contrarily, in a recent study, *Amilani et al. (2020)* used only the OIDP severity scores, because they found in a preliminary assessment that their participants' responses for the severity scale were more consistent compared to their responses for the frequency scale.

Previous researchers have also recommended use of frequency scales for other patient-reported outcome measures. *Chang et al. (2003)* evaluated the comparability of the 5-point intensity and frequency self-report rating scales for the questions from the Functional Assessment of Chronic Illness Therapy (FACIT)-Fatigue 13-item scale. Data from patients suffering from cancer, stroke, and HIV were calibrated separately to fit an item response theory-based rating scale model (RSM). The authors found no mean raw score difference between intensity and frequency rating scales, and a high correlation ($r = .86$, $p < .001$) between the intensity and frequency scores indicating their essential equivalence. The authors concluded that it was difficult to justify assessing both the response scales due to the essentially equivalent estimates they derived. However, they preferred the frequency response scaling as it provided fuller coverage of the fatigue continuum. *Hatt et al. (2017)* compared three different types of rating scales when administering potential questionnaire items to children with eye disorders and their parents. Children and parents in the study preferred the frequency scale over the difficulty and severity scales. In another study, *Krabbe & Forkmann (2012)* compared the usability of frequency and intensity terms as verbal anchors of self-report scales in patients suffering from a depressive episode. They found that frequency terms may have a slight advantage over intensity terms with regard to intraindividual stability across time so it might be advisable to use frequency terms when designing a self-report instrument that is intended to be applied in longitudinal assessments (*Krabbe & Forkmann, 2012*).

As mentioned before, there are advantages to using both frequency and severity scores; it helps minimize subjective clinical judgment, creates a wider and finer grading scale for assessment of oral diseases impacts, and enables clinicians and researchers to assess the impact of dental treatment on both symptom dimensions (*Elhai et al., 2006*). However, our study findings in conjunction with findings from previous studies mentioned above, suggest that both the scales provide redundant information and using one scale only can be considered to improve simplicity and efficiency. While the frequency response scale helps capture mild impacts that occur more frequently, the severity response scale helps capture more severe impacts that occur less frequently. Although more methodological research is needed to establish as to which response scale is better, currently there is more evidence to support the use of the frequency scale only in clinical and research settings.

## Strengths and limitations

The robustness of our study findings are supported by the use of a large sample of dental patients. We applied multiple analytical methodologies based on the classical methods and SEM at the scale and item levels to obtain a comprehensive view of the correlations between the frequency and severity response categories. Use of the SEM model allowed us to relate frequency and severity scales as latent variables while accounting for measurement

error (*Tarka, 2018*). We derived large correlations through the different analyses, which supports the preciseness of the study findings. Our study methodology can be applied to evaluating the response scales of other dPROMs which require patients to respond in terms of both frequency and severity.

In spite of the strengths, there were few limitations to our study. We acknowledge that the study findings are limited by the instrument (or dPROM) we chose to examine (OIDP). Although previous evidence suggests strong correlations among the OHRQoL instruments (*Robinson et al., 2003*; *John et al., 2016*), further research is needed to determine whether the findings from the current study would apply to other OHRQoL instruments. The correlation analyses in our study provide evidence of strong relationships between frequency and severity response scales, and the use of the frequency scale is recommended based on prior evidence (*Adulyanon & Sheiham, 1997*; *Åstrøm & Okullo, 2003*; *Chang et al., 2003*; *Krabbe & Forkmann, 2012*). Yet, the evidence regarding the preferred response scaling is mixed and more methodological work is needed to provide evidence that specifically supports the use of the frequency scale. For instance, Krabbe and Forkmann used a longitudinal study design, which supported the conclusion that frequency terms showed slightly greater stability across time. Change scores for the two OIDP scales should be compared as well as comparing the strength of the relationship of each to an external criterion.

## Significance of the study; recommendations for research and practice

The findings from our study support application of the OIDP frequency rating scale in clinical and research settings. The OIDP frequency scores had a strong correlation with the OIDP severity scores. Our findings align with previous research studies that have successfully adapted and applied the OIDP frequency scale to different cultural groups and contexts (*Adulyanon & Sheiham, 1997*; *Åstrøm & Okullo, 2003*; *Astrøm et al., 2005*; *Lawal, Taiwo & Arowojolu, 2013*; *Ostberg, Andersson & Hakeberg, 2008*; *Gülcan et al., 2014*; *Shekhawat, Samuel & Chauhan, 2021*). The evidence from our study supporting the use of OIDP frequency rating scale is of great practical value for clinicians. dPROMs like OIDP, with more than one type of rating scales are more time-consuming and burdensome for patients, as they require patients to recall the frequency as well as the severity of the perceived impact of an oral disease (*Reissmann, 2021*). Use of one response scale rather than two also will simplify the analysis and interpretation of the OIDP scores. Getting buy-in from dental providers and patients is necessary to capture dPROs more commonly and regularly (*John, 2018*; *Paulson et al., 2021*). Less time-consuming dPROMs with one response scale would be more appealing to clinicians who typically have several demands on their time.

Currently, there is no consensus on the most appropriate rating scale for the OIDP and our study offers promising evidence for applying the frequency impact format in clinical practice and research. OIDP correlates well with other OHRQoL questionnaires, that is, GOHAI and OHIP as they all measure the same construct (*Robinson et al., 2003*; *John et al., 2016*; *Oliveira & Nadanovsky, 2005*). OHIP and GOHAI, two of the most widely-used

OHRQoL instruments, also use the frequency impact format, attesting to the research and clinical community's interest and support for the frequency impact. Use of a frequency scale also would help to standardize assessment of outcomes across dental and medical treatments as patient-reported outcome measures used in medicine often employ frequency response scales (*Chang et al., 2003*; *Krabbe & Forkmann, 2012*).

Although further methodological work is needed, our study findings paves way for standardization of OHRQoL questionnaires and encourages the preferred use of frequency response scaling. Simple, short, and effective standardized questionnaires would make the sharing and interpretation of results from the OIDP (and other dPROMs) within the dental community more feasible (*Palaiologou & Kotsakis, 2020*). Regular assessment of dPROs would improve dentist-patient communications (*Palaiologou & Kotsakis, 2020*) and value-based oral health care (*Listl, 2019*), eventually advancing evidence-based dental practice.

## CONCLUSION

Our study findings and prior research support the use of the OIDP frequency scores only in clinical and research settings, as they are highly correlated with the OIDP severity scores. Use of only one response scale with OIDP would lower burden for patients and clinicians as well as researchers. The direct evidence in this study from a head-to-head comparison of frequency and severity oral impact format, indicated that severity is not superior compared to frequency, and that one of the most widely OHRQoL instruments can be successfully applied with only one response scale. The frequency impact format seems to be the preferred impact quality for OIDP, specifically, and for dPROMs in general.

### Funding
The National Institute of Dental and Craniofacial Research of the National Institutes of Health, USA, under Award Numbers R01DE022331 and R01DE028059, supported the study. The funders had no role in study design, data collection and analysis, decision to publish, or preparation of the manuscript.

### Grant Disclosures
The following grant information was disclosed by the authors:
The National Institute of Dental and Craniofacial Research of the National Institutes of Health, USA: R01DE022331, R01DE028059.

### Competing Interests
The authors declare there are no competing interests.

### Author Contributions
- Swaha Pattanaik conceived and designed the study, performed the experiments, prepared figures and/or tables, authored or reviewed drafts of the paper, and approved the final draft.

- Mike T. John conceived and designed the study, prepared figures and/or tables, authored or reviewed drafts of the paper, and approved the final draft.
- Seungwon Chung conceived and designed the study, performed the experiments, analyzed the data, prepared figures and/or tables, authored or reviewed drafts of the paper, and approved the final draft.
- San Keller conceived and designed the study, analyzed the data, authored or reviewed drafts of the paper, and approved the final draft.

## Ethics

The following information was supplied relating to ethical approvals (i.e., approving body and any reference numbers):

Institutional Review Board of the HealthPartners Institute in Minneapolis, MN.

## Data Availability

The raw data are available as Supplemental Files.

## Supplemental Information

Supplemental information for this article can be found online at http://dx.doi.org/10.7717/peerj.12717#supplemental-information.

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
