# Peer review of "Should the frequency, severity, or both response scales be used for multi-item dental patient-reported outcome measures?"

_PeerJ, doi:10.7717/peerj.12717_

## Round 0.1 · original submission · Major Revisions

Dear authors,
Thank you for submitting your manuscript to this prestigious journal.
Please follow the reviewers instruction
Best regards

·

Basic reporting

The English language use is clear, unambiguous, and professionally used throughout. Literature references are sufficient and context provided. Articles structure, figures, and tables are provided professionally. A relevant hypothesis has been phrased.

Experimental design

The primary research is within the Aims and Scope of the journal. The research question is well defined, relevant and meaningful. It is stated how research fills an identified knowledge gap. Methods are described with sufficient detail and information to replicate the study. The statistical power is excellent.

Validity of the findings

This paper adds relevant basic knowledge to the OHRQoL community. Conclusions are well stated, linked to the original research question & limited to supporting results. I congratulate the authors on this extraordinary research.

Additional comments

However, some minor points have to be addressed. A major number of references is missing in the reference list (i.e. John 2018, Paulson 2021, Mittal 2019, Reissmann 2021). Please correct. Furthermore, the given number of participants is misleading. Out of 2,115 participants, only 873 were included. This should be the number presented in the abstract.

Reviewer 2 ·

Basic reporting

The study investigated the correlation between frequency and severity rating scales of the OIDP instrument. Results demonstrated that the correlation estimates for all OIDP items were greater than 0.50. The estimated value of the correlation between the two total scores of frequency and severity is 0.85, which increases to 0.96 in the latent variable model. The authors concluded that OIDP frequency and severity scores are highly correlated, indicating that it is feasible to use only frequency scales in research and clinical settings.
In general, this work is very interesting and proposes a more effective way to conduct OTDP questionnaire surveys. The perspective of the study is very clear.
However, I have some concerns.
1.The patient's age, gender, symptoms, and other characteristics may affect their preference and performance for frequency and severity scores. Are there any differences in the average raw scores between the frequency and severity rating scales?
2. The Discussion section is important for a paper. The author enumerated many studies using frequency scale and talked about the importance of frequency scale. The comparison of severity scale and frequency scale should also be discussed.
3. Several references in the article cannot be found in the list, such as John, 2018; Paulson et al., 2021, Reissmann, 2021, John et al., 2016 (In Line 293).
This is a secondary research study using data from another cross-sectional study (Simancas-Pallares et al., 2020). Even this reference is not listed.
4. There are some grammatical errors, typing errors throughout the text. And it should be revised carefully to avoid any type of grammar or syntactical mistakes.
5. There are also some details problems. Such as *** Insert OIDP description Tables (Tables 1-3) here***.

Experimental design

Please see details in Basic reporting.

Validity of the findings

Please see details in Basic reporting.

---

## Round 0.2 · accepted · Accept

Dear authors,

Thank you for submitting your manuscript to this prestigious journal.
The article is now ready for publication.

Best regards

·

Basic reporting

I have no further recommendations.

Experimental design

I have no further recommendations.

Validity of the findings

I have no further recommendations.

Additional comments

I have no further recommendations.

Reviewer 2 ·

Basic reporting

The authors made sufficient revisions according to the modification comments and clarified the concerns raised by this reviewer. All the changes made to answer the reviewers help to improve the paper. And I suggest that this paper be accepted.

Experimental design

Please see the Basic reporting.

Validity of the findings

Please see the Basic reporting.